# Artificial-intelligence-driven discovery of catalyst *genes* with application to CO$_2$ activation on semiconductor oxides

Aliaksei Mazheika [1✉], Yang-Gang Wang[1,2], Rosendo Valero [3,4], Francesc Viñes [3], Francesc Illas [3], Luca M. Ghiringhelli[1,5], Sergey V. Levchenko[6✉] & Matthias Scheffler [1,5]

Catalytic-materials design requires predictive modeling of the interaction between catalyst and reactants. This is challenging due to the complexity and diversity of structure-property relationships across the chemical space. Here, we report a strategy for a rational design of catalytic materials using the artificial intelligence approach (AI) subgroup discovery. We identify catalyst *genes* (features) that correlate with mechanisms that trigger, facilitate, or hinder the activation of carbon dioxide (CO$_2$) towards a chemical conversion. The AI model is trained on first-principles data for a broad family of oxides. We demonstrate that surfaces of experimentally identified good catalysts consistently exhibit combinations of *genes* resulting in a strong elongation of a C-O bond. The same combinations of *genes* also minimize the OCO-angle, the previously proposed indicator of activation, albeit under the constraint that the Sabatier principle is satisfied. Based on these findings, we propose a set of new promising catalyst materials for CO$_2$ conversion.

[1] The NOMAD Laboratory at the Fritz-Haber-Institut der Max-Planck-Gesellschaft, 14195 Berlin-Dahlem, Germany. [2] Department of Chemistry and Guangdong Provincial Key Laboratory of Catalysis, Southern University of Science and Technology, 518055 Shenzhen, Guangdong, China. [3] Departament de Ciència de Materials i Química Física and Institut de Química Teòrica i Computacional (IQTCUB), Universitat de Barcelona, c/ Martí i Franquès 1, Barcelona 08028, Spain. [4] Zhejiang Huayou Cobalt Co.,Ltd., No. 18 Wuzhen East Road, Tongxiang Economic Development Zone, 314500 Jiaxing, Zhejiang, China. [5] The NOMAD Laboratory at the Humboldt University of Berlin, 12489 Berlin, Germany. [6] Skolkovo Institute of Science and Technology, Skolkovo Innovation Center, Bolshoy Boulevard 30, bld. 1, 121205 Moscow, Russia. ✉email: alex.mazheika@gmail.com; s.levchenko@skoltech.ru

The need for converting stable molecules such as carbon dioxide ($CO_2$), methane, or water into useful chemicals and fuels is growing quickly along with the depletion of fossil-fuel reserves and the pollution of the environment[1–3]. Such a conversion does not have a satisfactory solution, so far. In particular, $CO_2$ conversion remains one of the most important societal and technological challenges[1,2,4–8].

The general understanding in heterogeneous catalysis is that a stable molecule such as $CO_2$ needs to be "prepared" before its catalytic conversion occurs. This leads to the notion of molecular activation[9]. However, on one hand, this notion encompasses a very wide variety of processes (adsorption, photo-excitation, application of electric field, etc.) and materials (including compositional and structural variability), and it remains unclear which properties of the catalytic material and the adsorbed molecule determine the final chemistry, what is the relationship between the two sets of properties, and how general this relationship may be. On the other hand, finding the set of descriptive parameters of a catalytic material that characterize the catalytic performance in a particular process, or even in general for a given reactant, would be very valuable, because it would allow us to quickly search for promising candidate catalysts using rational design[10–17]. We call these properties materials *genes*. The *genes* do not necessarily correlate with catalytic activity by themselves. Similar to biological genes, their role depends on the combination in which they occur, and can be either beneficial or detrimental to the catalytic activity.

Several strategies exist to find such properties for a given reaction. One way is to explore the free-energy surface for each catalyst candidate, which is a slow and resource-consuming process, and currently computationally unfeasible for many materials on a high-throughput basis. An alternative approach consists in searching for a correlation between experimentally determined material's properties and its catalytic performance. Such a strategy requires consistent experimental measurements at well-defined conditions for a set of materials. To the best of our knowledge, such consistent data have not been reported so far for $CO_2$ conversion on semiconductor oxides. Moreover, available publications usually do not report unsuccessful experimental results. These issues and a strategy to address them have been recently discussed in our publication[18].

Yet another strategy is to find an indicator of activation, namely, a property of the system that directly indicates the certain catalytic performance of the material[10]. Indicators are distinguished from materials *genes* based on a qualitatively different level of computational complexity. The indicator can still be unfeasible or hard for a high-throughput study of hundreds of thousands or millions of materials. However, when it can be calculated for a few tens or hundreds of materials in a reasonable time, these data can then be used to find materials *genes* that control the value of the indicator. Since a direct search for a relationship between the indicator and catalytic performance of material would also require a consistent set of data of turnover frequency (TOF), selectivity, and yield values, one could instead consider several most promising indicators, find out which materials are good catalysts, and then check which indicators correlate with this observation. This approach also addresses the problem of defining activation in terms of the adsorbed-molecule properties as potential indicators of catalytic activity.

Catalytic conversion of $CO_2$ requires activation of other reactants as well, e.g., molecular hydrogen, water, or methane. In particular, hydrogen can serve as an environmentally friendly reagent that can be produced by water electrolysis or photo-splitting avoiding extra $CO_2$ emissions[19–21]. Also, oxygen vacancies have been proposed as active sites for $CO_2$ conversion on some materials[22]. Therefore, predictions of catalytic activity of materials for $CO_2$ conversion can be refined based on analysis of activation of other reactants and defects. An additional challenge is to ensure that the useful products, as well as the surface catalytic activity, are preserved under the conditions of activation and subsequent conversion. While the strong C–O double bonds in $CO_2$ can be weakened or even broken by adsorption at a solid surface at an elevated temperature, this may also lead to too strong adsorption or further dissociation of the molecule, so that the catalytic surface is poisoned by carbonate or carbon deposits. Weak adsorption, on the other hand, means no activation.

In this work, we combine first-principles calculations with an artificial-intelligence (AI) method, subgroup discovery (SGD), to identify pristine materials properties that optimize indicators of catalytic $CO_2$ activation. Moreover, SGD allows identifying one or more distinct combinations of materials features (*genes*) that promote activation. We focus on oxide materials as candidate catalysts. Oxides are structurally and compositionally stable under realistic temperatures and can be less expensive than the traditional precious metal-containing catalysts[23–25]. Activation of other reactants and defects are not considered. As shown below, meaningful predictions can be made based solely on the analysis of the adsorption properties of $CO_2$ on pristine surfaces. This confirms that these properties are good indicators of activation with a viable optimization pathway at least for the chosen class of materials. The Sabatier principle is taken into account by ensuring that the adsorption energy is not too large or too small. In order to ensure reproducibility of our AI data analysis, we provide all necessary metadata (input parameters) and workflow in the easily accessible form of a Jupyter notebook[26]. We argue that, with the ever-growing importance and complexity of AI, such detailed and tutorial documentation is a necessity of good scientific practice. Our approach is applicable to a wider class of materials and molecules, not limited to oxides or $CO_2$. Our study by no means encompasses all possible mechanisms of $CO_2$ conversion on oxide surfaces, but it offers a clear design path among many possible ones.

## Results

**$CO_2$ activation**. We find that on semiconductor oxide surfaces $CO_2$ is chemisorbed exclusively when the carbon atom binds to surface O-atoms. All other minima of the potential-energy surface are found to be either metastable or correspond to physisorption. Therefore, there are as many different potential chemisorption sites as there are unique O-atoms at the surface. The dataset includes all non-equivalent surface O-atoms on the 141 considered surfaces of 71 materials, which sum up to 255 unique adsorption sites. Among these sites on about 4% (10 out of 255) $CO_2$ prefers to physisorb, i.e., any chemisorbed state is metastable with respect to the physisorbed one. The physisorption can be easily identified by an almost linear geometry of the adsorbed molecule, and a C–O bond distance very close to the C–O bond length in a gas-phase $CO_2$ molecule, 1.17 Å.

We considered six different candidate indicators of $CO_2$ activation, including OCO-angle and C–O bond distance. The bending of the OCO-angle in the adsorbed $CO_2$ molecule relative to the gas-phase value of 180° (linear configuration) has been previously proposed[27] and is widely accepted as a good indicator of activation. For gas-phase $CO_2$, it is understood that the C–O double bond is weakened when an electron is added to the lowest unoccupied orbital, because it is of antibonding ($\pi^*$) character with a concomitant bending of the molecule. There is a one-to-one mapping between the C–O bond length $l$(C–O) and the OCO-angle in gas-phase $CO_2^{\delta-}$ for a range of $\delta > 0$ (red curve in Fig. 1). However, this is not the case for the adsorbed $CO_2$ (dots in Fig. 1). There is a subset of adsorbed $CO_2$ that is close to the

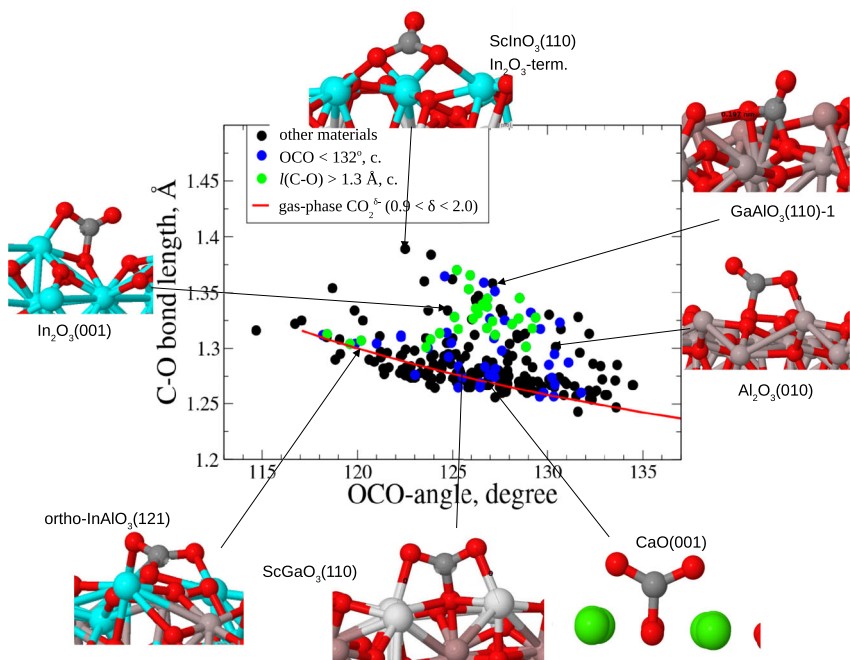

**Fig. 1 Correlation between the larger of the two C–O bond lengths and the OCO-angle for charged gas-phase and adsorbed CO2.** The OCO-angle in charged gas-phase $CO_2$ is shown with the red line, and adsorbed $CO_2$ structures are shown with the dots. Colored dots: blue—adsorption sites from the unconstrained subgroup with OCO < 132°, green—subgroup of sites with $l$(C–O) > 1.30 Å, black—the remaining samples (see the text). The subgroups obtained with Sabatier principle constraint are marked with "c".

red line, but there are many cases where $l$(C–O) is substantially larger for a given OCO-angle. This is in contrast to metal alloy nanoparticle catalysts, where there is a better correlation between OCO-angle and $l$(C–O)[28]. Also, a longer C–O bond reflects a weakening and readiness for further chemical transformations. Thus, the bond elongation itself may be an alternative indicator of activation. A look at the adsorbed $CO_2$ structures reveals that, on sites following the gas-phase correlation, the molecule adsorbs in nearly symmetric adsorption structures with nearly equal length of the two C–O bonds. In the other cases one O-atom of $CO_2$ is close to surface cation(s), leading to a pronounced asymmetry of the adsorbed molecule.

Other considered potential indicators of activation include Hirshfeld charge[29] of adsorbed $CO_2$ (a direct indicator of the charge transferred to $CO_2$), the dipole moment of the surface along the surface normal per adsorbed $CO_2$ molecule (includes charge transfer to the molecule, as well as adsorption induced surface relaxation), the difference in Hirshfeld charges of C and O-atoms in an adsorbed $CO_2$ molecule (indicates the ionicity of C–O bonds), and the difference in Hirshfeld charges of the O-atoms in the adsorbed molecule (indicates asymmetry of the adsorbed molecule)[9,29].

**Subgroup discovery.** To find out which properties (features) of the clean surfaces determine when a given activation indicator is maximized or minimized, we employ the subgroup-discovery (SGD) approach[30–34]. Given a dataset and a target property known for all data points, the SGD algorithm identifies subgroups with "outstanding characteristics" (see further for the criteria for being outstanding) and describes them by means of conjunction of basic propositions (selectors) of the kind "($f_1 < a$) AND ($f_2 \geq b$) AND ...", where $f_i$ is a feature and $a$, $b$ are threshold values also found by SGD. In the framework of SGD, we call the selected primary features $\{f_1, f_2, ...\}$ materials *genes*. Thus, SGD identifies both the outstanding subgroups and the relevant materials *genes* for a given target property.

Obviously, the selectors should only contain features that are much easier to evaluate than the target property. In the presented work, the considered features include properties of gas-phase atoms that build the material, and properties of the pristine material (properties of the bulk phase and of the pristine relaxed surface). Overall 46 primary features have been considered. The full list is presented Supplementary Table 3. Our strategy is to provide an almost exhaustive list of features, and use data analytics to select materials *genes* from this list. Some of these features have been explored previously as descriptors of catalytic activity for semi-conducting and metallic oxides[35–38]. O $2p$-band center features have been shown to correlate with catalytic properties of both semiconducting and metallic oxides[35,37]. In particular, most of the features (or closely related ones) mentioned in ref. [36], inspired by the work of Grasselli[39], are included in our set, except oxygen vacancy formation energy, which is relevant for the oxidation catalysis, while here we are interested in partial or complete reduction. Additional important features in our work (see below) include features related to the polarizability of surface cations, which describe the long-range surface response to charged adsorbates. A subset of features from our list has been recently used successfully for predicting catalytic properties of metallic oxides[38], along with additional features relevant specifically for metallic oxides (such as partial electronic state fillings).

The features selected by the SGD are summarized in Table 1.

The outstanding subgroup should satisfy several criteria. It should be statistically relevant; therefore the subgroups of too small size should be penalized. Target-property values (OCO-angle, C–O bond length, etc.) for subgroup samples should be as different as possible from corresponding gas-phase values since their change upon adsorption indicates $CO_2$ activation[33]. To achieve this, two requirements are imposed simultaneously: (i) The target-property values for subgroup members should be smaller or larger (depending on the target) than a certain value (a cutoff), and (ii) the target-property values are minimized or maximized within the cutoff. The latter condition gives

**Table 1 Features that appear in the top SGD selectors (see text).**

| symbol | Meaning |
|---|---|
| $IP_{min/max}$ | Ionization potential, minimal and maximal in the pair of atoms A and B; calculated as $E_{atom} - E_{cation}$ |
| $EA_{min/max}$ | Electron affinity, minimal and maximal in the pair of atoms A and B; calculated as $E_{anion} - E_{atom}$ |
| $EN_{min/max}$ | Mulliken electronegativity, minimal and maximal in the pair of gas-phase atoms A and B |
| $r_{-1}^{min}$, $r_{-1}^{max}$ | Radii of the maximum value of the Kohn-Sham radial wave functions of the spin-unpolarized spherically symmetric atom for HOMO-1, maximum (max) and minimum (min) in the pair of atoms A and B |
| $r_{+1}^{min}$, $r_{+1}^{max}$ | Radii of the maximum value of the Kohn-Sham radial wave functions of the spin-unpolarized spherically symmetric atom for LUMO, maximum (max) and minimum (min) in the pair of atoms A and B |
| $M$ | Energy at which the surface O 2$p$-band projected density of states (PDOS) is maximal |
| $d_1$, $d_2$, $d_3$ | Distances from surface O-atom to the first-, second-, and third-nearest cations |
| $W$ | Work function $W$, as the negative of the valence-band maximum ($W = -VBM$) with respect to vacuum level |
| $q_{min}$, $q_{max}$ | Minimal and maximal Hirshfeld charges of cations in the pair A and B, calculated as an average for all surface cations of a given type |
| $\Delta$ | Bandgap |
| $CBM$ | Conduction band minimum |
| $Q_5$, $Q_6$ | Local-order parameter with $l = 5$ or 6 |
| $PC$ | Weighted surface O 2$p$-band center |
| $\alpha_O$, $C_6^O$ | Polarizability and $C_6$-coefficient for surface O-atom obtained from many-body dispersion scheme |
| $\alpha_{min}$, $\alpha_{max}$, $C_6^{min}$, $C_6^{max}$ | Polarizability and $C_6$-coefficient for cations, minimal and maximal in the pair A and B, calculated as an average for all surface cations of a given type |
| $q_O$ | Hirshfeld charge of O-atom at the surface |
| $wid$ | Square root of the second moment of surface O 2$p$-band |
| $wid_{min}$, $wid_{maxS}$ | Square root of the second moment of PDOS of cations within valence-band, minimal and maximal in the pair A and B, calculated as an average for all surface cations of a given type |
| $c_{min}$, $c_{max}$ | First moment for PDOS of cation within valence-band, minimal and maximal in the pair A and B, calculated as an average for all surface cations of a given type |
| $\varphi_{1.4}$, $\varphi_{2.6}$, $\varphi_{1.4} - \varphi_{2.6}$ | Electrostatic potentials above surface O-atom at 1.4 and 2.6 Å and their difference. 1.4 Å corresponds to the average length of the bond between C and surface O, 2.6 Å is the minimal distance from surface O to C-atom of physisorbed carbon-dioxide molecule as observed from our calculations |
| $L_{min}$, $L_{max}$ | Energy of lowest unoccupied projected eigenstate of surface cations, minimal and maximal in the pair A and B, calculated as an average for all surface cations of a given type |
| $kurt$ | Kurtosis of surface O 2$p$-band PDOS |
| $U$ | Eigenstate with least negative value in surface O 2$p$-band |
| $BV$ | Bond-valence value of surface O-atom |

preference to subgroups with smaller or larger target-property values among similarly sized subgroups within the cutoff. The value of the cutoff is a parameter. As it approaches the optimal value of an activation indicator among all data points, additional or alternative materials *genes* and their combinations leading to stronger activation are identified. We explore the whole range of the parameter for each target property (for OCO-angle—123°, 124°, 126°, 128°, 130°, and 132°; for $l$(C–O)—1.26 Å, 1.28 Å, and 1.30 Å).

In addition to these criteria, we consider the requirement that adsorption energies are not too strong and not too weak for most of the samples in a subgroup. Strong activation (i.e., strong weakening of the C–O bonds) can be achieved by strong binding to the surface. It is well known that good catalytic performance requires a balanced adsorption strength. This is known as Sabatier principle. In addition to the practical value of identifying subgroups that satisfy this principle, comparison of subgroup selectors obtained with and without this requirement helps to identify combinations of materials features that promote desired changes in target properties and at the same time yield intermediate adsorption energies.

Sabatier principle is reflected by a characteristic volcano-type behavior of catalytic activity as a function of adsorption energy of reactants and intermediates. The position of the top of the volcano depends on particular reactions and conditions. It can be estimated from condition $|\Delta G| \sim 0$, where $\Delta G$ is the Gibbs free energy of adsorption. For $CO_2$ adsorption at room temperature and partial $CO_2$ pressure of 1 atm this condition corresponds to about $-0.5$ eV adsorption energy[40]. At temperatures around 450 °C (typical conditions for $CO_2$ methanation[41]) $\Delta G = 0$

corresponds to adsorption energy $-1.7$ eV[41]. Therefore, for catalytic conversion at low or moderate temperatures this implies that $CO_2$ adsorption energies should be in the range from between $-2.0$ and $-0.5$ eV.

These requirements are implemented in the following quality functions that are maximized during the search for subgroups. In particular, for OCO-angle minimization we use:

$$F(Z) = \theta_{cut}\left[\frac{s(Z)}{s(Y)} \cdot \left(\frac{\max(Z) - \alpha_g}{\min(Y) - \alpha_g}\right) \cdot u(p)\right] \quad (1)$$

and for C–O bond maximization the following quality function was applied:

$$F(Z) = \theta_{cut}\left[\frac{s(Z)}{s(Y)} \cdot \left(\frac{\min(Z) - l_g}{\max(Y) - l_g}\right) \cdot u(p)\right] \quad (2)$$

where $Y$ is the whole dataset, $Z$—a subgroup, $s$—size (number of data points), min and max – minimal or maximal value of the target property, $\alpha_g$ and $l_g$ are the gas-phase values of OCO-angle and C–O bond distance, 180° and 1.17 Å, respectively, and $\theta_{cut}$ is the Heaviside step function which is equal 1 if all data points in the subgroup satisfy the cutoff condition and 0 otherwise. Thus, larger values of the quality function $F(Z)$ are obtained for those subgroups in which minimal (maximal) value of a target property is close to the maximal (minimal) value of the whole sampling with respect to the gas-phase value of $CO_2$ molecule. The use of maximum/minimum instead of a median is done to ensure that a target property is optimal for as many members of a subgroup as possible. The gas-phase reference values are usually significantly

different from the "chemisorption" subset. Therefore, the term in squared brackets in Eqs. (1) and (2) can noticeably contribute only when the sizes of candidate subgroups are similar.

The term $u(p)$ in Eqs. (1) and (2) is added in order to account for Sabatier principle in SGD framework. We have implemented a multitask quality function, where a factor $u(p)$ increases the quality of subgroups with adsorption energies falling within this range. This is formulated in terms of the information gain[34], i.e., reduction of the normalized Shannon entropy. We perform the SGD for each target property both explicitly accounting for the Sabatier principle and without it. The latter case is equal to $u(p) = 1$ in Eqs. (1) and (2)[34].

We note that SGD is qualitatively different from machine-learning classification/regression techniques such as neural networks, kernel regression methods, or decision-tree regression (DTR[42]) (e.g., random forest). SGD is typically referred to as a supervised descriptive rule-induction technique[43], i.e., it uses the labels assigned to the data points (the values of the target property) in order to identify patterns in the data distribution (the statistically exceptional data groups) and the rules defining them (the selectors), by optimizing a quality function which is a function of the distribution of values of the target property[43]. While there are apparent similarities between SGD and DTR as both methods yield models in terms of physically interpretable selectors (usually, inequalities) on a selected subset of the input features, the analogy stops at this level, as SGD focuses at (and only at) subgroups from the very beginning and says nothing about the data that are not in the subgroup. In contrast, DTR determines a global partitioning of the input space by minimizing a global quality function, i.e., the quality of a single subset is secondary with respect to the resulting quality of all subsets partitioning the whole dataset. In other words, for finding distinct combinations of materials genes driving desirable changes in a particular target property (possibly different combinations leading to the same result), the SGD approach has significantly higher flexibility and reliability. This is demonstrated below for a DTR analysis for our target properties.

The metadata and workflow for the AI analysis are documented in the Jupyter notebook[26].

**Results of the subgroup discovery**. The SGD for OCO-angles was done with Eq. (1) for the quality function, and OCO as a target property, since smaller angles indicate larger charge transferred to the molecular $\pi^*$ orbital. The subgroup selectors obtained with different OCO-angle cutoffs (126°, 128°, 130°, and 132°) with or without the adsorption energy constraint are listed in Table 2 (for more details see the Supplementary Table 4). Analysis of these subgroups reveals that the angle reduction is determined by an interplay of several factors: an electron transfer from the cations to surface O-atoms, delocalization of electron density between cations and O-atoms, and coordination of the surface O-atoms. Without the Sabatier principle constraint, the OCO-angle reduction below 132° is mainly due to the electron accumulation at the O-atom of the clean surface. This is expressed by the conditions of more negative Hirshfeld charge on O-atoms ($q_O < \ldots$), not very low IP of at least one cation ($IP_{max} > \ldots$), and increased polarizability of the surface O-atom on which $CO_2$ is adsorbed ($C_6^O > \ldots$). Upon adsorption of $CO_2$, this charge on the surface O-atom is readily available for transfer to $CO_2$. When the Sabatier principle constraint is introduced, the OCO < 132° subgroup also includes sites with a pronounced electron transfer to $CO_2$, but with a lower-energy O $2p$-band maximum ($M < \ldots$) with respect to vacuum level, and a larger kurtosis ($kurt > \ldots$). These conditions imply reduced inter-electronic repulsion around the surface O-atom achieved by partial delocalization of the charge density.

At lower OCO cutoffs, the subgroup selectors include coordination descriptors $Q_i$, $i = 5, 6$. Without Sabatier principle, sites with larger $Q_i$ are selected, and vice versa. Larger $Q_i$ indicates lower coordination of the O-atom. This reduces electron repulsion and therefore facilitates electron transfer to the O-atom of the clean surface. However, this also increases the bonding strength of $CO_2$ to the surface. This explains why selectors of subgroups obtained with Sabatier principle include the opposite conditions ($Q_5 < \ldots$).

Other surface features describing electron distribution are related to Madelung potential: electrostatic potential and field ($\varphi_{1.4}$, $\varphi_{2.6}$, and $\Delta\varphi = \varphi_{1.4} - \varphi_{2.6}$) and distances between the O-atom and surface cations. More open surface structure with larger distances between cations at the O site facilitates charge transfer to adsorbed $CO_2$ molecule, since the Madelung potential from the nearby cations is reduced. This is reflected in the appearance of propositions involving features $d_1$, $d_2$, and $d_3$. For example, for the OCO ≤ 130° subgroups, imposing energy constraint changes proposition ($d_1 > \ldots$) to ($d_1 < \ldots$), which implies an increased energy cost for transferring electrons to $CO_2$. Larger electric fields $\Delta\varphi$ around the adsorption site imply stronger localization of electron density on O-atoms, and thus also improve the efficiency of charge transfer to the adsorbed molecule.

The smaller OCO subgroups with Sabatier principle also include propositions implying increased polarizability of both cations ($C_6^{min} > \ldots$). Another support-defining condition is that the radius of the lowest unoccupied orbital for the metal atoms should not be small ($r_{+1} \geq \ldots$). This requirement is true for most cations with negative electron affinities (Supplementary Fig. 4). Analysis of adsorbed $CO_2$ structures and Hirshfeld charges reveals that this condition together with the higher polarizability of cations at the *pristine* surface encompasses two scenarios: (i) additional electron transfer to $CO_2$ upon adsorption and (ii) stronger binding between O-atoms in $CO_2$ and surface cations. When scenario (ii) dominates, $CO_3^{\delta-}$ anion lies nearly horizontally at the surface, and is bound with nearby cations by chemical bonds via its oxygen atoms. Such a structure leads to small OCO-angles in $CO_3^{\delta-}$ (around 120°), even if charge transfer is limited. Thus, increased bending of adsorbed $CO_2$ occurs due to charge transfer over larger distances and/or distortion of the adsorbed molecule and the surface, both leading to weaker adsorption. The cases where both scenarios are active include the same sites as in the subgroups with elongated $l$(C–O), as described below.

In order to obtain the subgroups of adsorption sites with larger $l$(C–O), we performed the SGD with the quality function Eq. (2) and $l$(C–O) as target property. The results for $l$(C–O) cutoffs 1.26, 1.28, and 1.30 Å are summarized in Table 2 and Supplementary Table 5. In contrast to OCO, the analysis of the obtained top subgroups shows a much less pronounced or no effect of imposing Sabatier principle on the distribution of adsorption energies within the subgroups. This is because sites with too strong adsorption are excluded based on $l$(C–O) threshold alone, without the need to introduce the energy constraint. For example, the range of $l$(C–O) for the top $l$(C–O) > 1.26 Å subgroup without constraining adsorption energies is the same as for the top OCO < 130° subgroup, but it contains significantly more sites with intermediate adsorption energies.

Electron transfer to an adsorbed $CO_2$ molecule increases both the OCO bending and C–O bond elongation. The main difference between OCO and $l$(C–O) subgroups is that in the latter an additional mechanism of increasing $l$(C–O) is in effect, namely a covalent bonding between one O-atom of the $CO_2$ molecule and the nearest surface cation. This can be concluded from the analysis of adsorption geometries, and correlates with the

**Table 2 Top subgroups and their selectors obtained by minimization of OCO-angle and maximization of $l$(C–O) with/out Sabatier principle (energies are in eV, distances are in Å, charges are in units of absolute electron charge, polarizabilities are in Bohr$^3$).**

| cutoff | size | selector | cutoff | size | selector |
|---|---|---|---|---|---|
| OCO minimization without Sabatier principle constraint | | | OCO minimization with Sabatier principle constraint | | |
| 126 | 19 | $L_{max} > -2.70$ ($L_{min} > -2.19$, $CBM > -3.40$, $r_{+1}^{max} \leq 2.83$, $W < 5.80$, $U > -5.61$) $IP_{max} \geq -6.05$ $\alpha_{max} \leq 184.5$ $\Delta\varphi > 1.33$ $q_{max} \leq 0.59$ $wid \leq 1.59$ $wid \geq 0.58$ | 126 | 15 | $L_{min} \geq -5.1085$ $\varphi_{2.6} \geq 0.3033$ $\Delta\varphi \leq 1.0622$ ($c_{max} \leq -8.5915$) $d_1 \geq 1.82$ $d_2 \geq 2.005$ $r_{+1}^{max} > 2.83$ |
| 128 | 44 | $EA_{max} \geq -0.43$ $Q_6 \geq 0.51$ $\alpha_{max} \geq 50.4$ ($C_6^{max} \geq 389.5$, $\alpha_O \leq 2.70$) $\Delta\varphi \geq 1.00$ $q_{min} \leq 0.49$ | 128 | 30 | $C_6^{min} \geq 369.5$ $L_{max} \geq -4.73$ ($r_{+1}^{min} \leq 2.82$, $IP_{min} \leq -5.83$, $r_{HOMO}^{min} \leq 1.41$) $Q_5 \leq 0.83$ $\Delta\varphi \geq 0.60$ $r_{+1}^{max} \geq 2.80$ $C_6^O \leq 12.10$ |
| 130 | 77 | $L_{max} \geq -5.23$ $EA_{max} \leq 0.16$ ($C_6^{max} \geq 389.5$, $IP_{max} \geq -7.00$) $d_1 \geq 1.82$ $d_2 > 2.10$ | 130 | 40 | $\varphi_{2.6} \geq -0.15$ $\Delta\varphi \geq 0.73$ $d_1 \leq 2.01$ $d_2 \geq 1.96$ $d_3 \geq 2.025$ ($c_{min} \leq -9.07$, $W \geq 5.10$) $q_{min} \leq 0.49$ $r_{+1}^{min} \geq 1.94$ |
| 132 | 139 | $IP_{max} \geq -6.99$ $q_O \leq -0.32$ $C_6^O \geq 10.36$ | 132 | 58 | $q_O \leq -0.3386$ $M \leq -6.292$ $kurt \geq 2.1035$ $IP_{max} \geq -6.2085$ $r_{HOMO}^{min} \leq 1.407$ ($IP_{min} \leq -5.91$, $r_{+1}^{min} \leq 2.82$) |
| $l$(C–O) maximization without Sabatier principle constraint | | | $l$(C–O) maximization with Sabatier principle constraint | | |
| 1.26 | 121 | $C_6^{min} \geq 343.5$ $\varphi_{2.6} \leq 0.66$ $Q_5 \leq 0.83$ $M \geq -8.05$ ($PC \geq -9.32$) | 1.26 | 56 | $CBM \geq -5.17$ ($L_{min} \geq -5.11$) $\Delta\varphi \leq 1.13$ $PC \geq -8.62$ $d_3 \leq 2.48$ $M \leq -6.06$ |
| 1.28 | 38 | $EA_{max} \leq 0.005$ $d_2 > 2.22$ $M \leq -4.12$ | 1.28 | 30 | $W \geq 5.10$ ($M \leq -5.19$, $U \leq -4.92$, $PC \leq -7.21$) $d_2 > 2.14$ $q_{min} < 0.48$ |
| 1.30 | 27 | $U \leq -5.34$ $d_2 > 2.14$ $q_{min} < 0.48$ $kurt \geq 2.10$ ($q_{max} \geq 0.47$) | 1.30 | 27 | $EA_{max} \leq 0.005$ ($W \geq 5.10$, $M \leq -5.19$, $U \leq -4.92$, $PC \leq -7.21$) $EN_{min} \leq -3.19$ ($W \geq 5.10$, $q_O \geq -0.45$, $c_{max} \leq -7.18$, $r_{HOMO}^{min} \leq 1.41$, $\varphi_{1.4} \leq 2.40$, $c_{min} \leq -8.135$, $q_{max} \geq 0.47$, $M \leq -5.19$, $IP_{min} \leq -5.91$, $wid \geq 0.58$, $U \leq -4.92$, $r_{-1}^{max} \geq 0.97$, $PC \leq -7.21$, $\Delta\varphi \leq 1.81$) $d_2 > 2.14$ $q_{min} < 0.48$ $kurt \geq 2.51$ |

Proposition replacements that do not change the support are shown in parentheses.

presence of proposition ($EA_{max} \leq 0.005$ eV), selecting cation species that can accept electron density, e.g., from an O-atom in adsorbed $CO_2$ molecule. Other proposition that appears in most selectors of top subgroups is ($d_2 > 2.14$ Å) or ($d_2 > 2.22$ Å)—larger distances to the second nearest cation from an O-atom. Larger elongation of the C–O bond is achieved by the asymmetry of the cation types at the surface, where one can bind an O-atom of the adsorbed $CO_2$, while the other (located further away) cannot. An example asymmetric $CO_2$ adsorption structure is shown in Supplementary Fig. 5.

Other propositions indicate a moderate charge transfer to adsorbed $CO_2$ molecule as in the case of OCO subgroups with adsorption energy constraint. Propositions ($M \geq -8.05$ eV), ($PC \geq -9.32$ eV) in $l$(C–O) < 1.26 Å subgroups imply enhanced charge density on the surface O-atoms, since electron–electron repulsion raises energies of O 2p-band states. However, at larger $l$(C–O) cutoffs the electron transfer is balanced by such propositions as

($M \leq -5.19$ eV), ($U \leq -4.92$ eV), and ($W \geq 5.10$ eV) indicating limited electron transfer. These propositions point to more covalent bonding between cations and surface O-atom. Rather persistent proposition observed in many selectors of $l$(C–O) subgroups is the limit of minimal charge on surface cations ($q_{min} < 0.48e$). It also shows the limitation of the charge transfer from one type of cations to surface oxygen atoms.

In general, we find that subgroups obtained with smaller cutoffs do not have a strong overlap with subgroups with larger cutoffs for OCO. In particular, for subgroups with close cutoffs the overlap can be smaller than 50% of the smaller subgroup (but is never below 30%). Interestingly, for $l$(C–O) the situation is opposite: subgroups with tighter cutoffs are mostly contained in the subgroups for more relaxed constraints. This means that, while larger values of $l$(C–O) are mainly controlled by the same or additional *genes*, smaller values of OCO are due to alternative *genes*. The overlap of OCO subgroups becomes even smaller

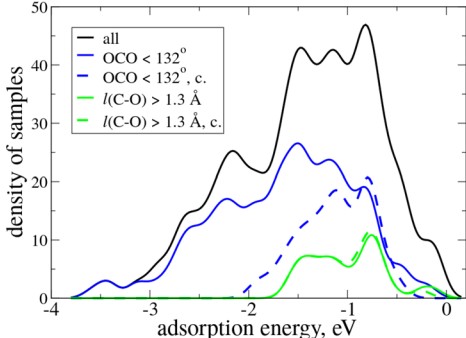
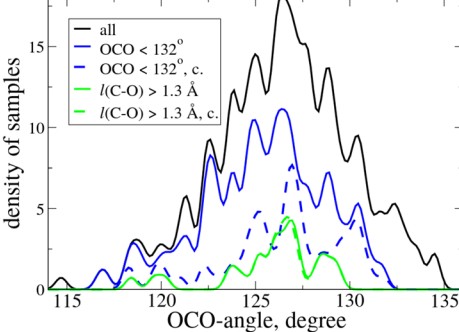

**Fig. 2 Distribution of adsorption energies (left) and OCO-angles (right).** The distribution is shown for the whole dataset (black), for the top subgroups of sites with OCO < 132° angles (blue) and $l$(C–O) > 1.30 Å (green). The subgroups obtained with adsorption energy constraint are marked with "c." and shown with dashed lines. The adsorption energy $E_{ads}$ is defined as the difference between the total energy of the slab with adsorbed $CO_2$ and the sum of total energies of the clean slab and an isolated $CO_2$ molecule.

when Sabatier principle is included, confirming the absence of a universal mechanism for OCO-angle reduction that is compatible with moderate adsorption energy.

In summary, we find that, while an increased electron density at the O adsorption site is necessary for chemisorption and leads to both OCO bending and C–O bond elongation in an adsorbed $CO_2$ molecule, there are additional actuators for these effects that are different for different target properties. The OCO-angle is in general minimized by increasing electron transfer to the O site. However, this also leads to strong adsorption for many materials (Fig. 2). To satisfy Sabatier principle, the electron transfer to $CO_2$ must be moderate. This is achieved by delocalization of charge density around O sites and/or by distortion of the adsorbed molecule due to the formation of covalent bonds between O-atoms in $CO_2$ and surface cations. The largest C–O bond elongations are achieved when both charge transfer to adsorbed $CO_2$ and the covalent interaction are present, and local geometry around surface O-atom provides the asymmetry in adsorption structure. This mechanism automatically fulfills the Sabatier principle.

The subgroups found by SGD for the dipole moment induced by $CO_2$ adsorption, its total Hirshfeld charge, and the difference of charges on C and O-atoms significantly overlap with the subgroup of smaller OCO-angles. The subgroup found by maximizing the difference of Hirshfeld charges on O-atoms of an adsorbed $CO_2$ largely overlaps with the subgroup of sites delivering larger $l$(C–O). In general, these indicators are not better than OCO or $l$(C–O). Therefore, below we focus on OCO-angle and $l$(C–O) as indicators of $CO_2$ activation. More details about the other indicators can be found in Supplementary Discussion.

**Comparison with experimental results**. To address the question which of the discussed properties can serve as an indicator of the catalytic activity, we compare our predictions to reported experimental results (Table 3). It should be stressed that the available experimental data are scarce, and results are difficult to compare quantitatively. We consider thermally and, for completeness, some photo-driven catalysis and thus also include supported metal catalysts with the considered oxides as support. Despite possibly different mechanisms for $CO_2$ conversion in the different types of catalysis, we believe that the properties of adsorbed $CO_2$ molecule can still serve as indicators of catalytic activity. Thus, it is possible that under such a daunting situation a reliable indicator of $CO_2$ activation can still be identified. As described below, our analysis confirms this hope.

First, we consider materials with the sites from subgroups obtained by minimization of OCO-angle without Sabatier principle constraint[27]. For quite many materials from these subgroups, independent of the cutoff value, there are no reports of successful $CO_2$ conversion, even when they are used as supports for metal nanoparticles (Table 3). This is explained by the fact that absolute adsorption energies for these materials are above 2 eV (Fig. 2 left, Supplementary Table 4), indicating that their surfaces will be permanently poisoned by carbonate species at low or intermediate temperatures. This means that on materials with these sites hardly any reaction of $CO_2$ conversion can proceed at low, especially room temperature. Moreover, as shown in Table 3, even at increased temperatures, 700–750 °C, the activity of these materials is low. Some of them have been considered as candidates for carbon capture and storage (CaO, SrO, BaO, and $Na_2O$)[44], which implies the formation of stable carbonates rather than $CO_2$ transformation. Thus, we conclude that OCO-angle alone is not a good indicator of enhanced catalytic activity in $CO_2$ conversion.

On the other hand, several of the materials with sites from $l$(C–O) > 1.30 Å subgroups (independent on either with or without Sabatier principle constraint) are known as good materials for $CO_2$ conversion (Table 3) in different reactions proceeding at room or higher temperatures. For these sites, the absolute adsorption energies already satisfy the Sabatier principle (Fig. 2, left), as discussed above. We note that, contrary to what one may expect, there is no correlation between the adsorption energy and the value of $l$(C–O) (see Supplementary Fig. 5). Although there is a general trend, there are also significant variations in $l$(C–O) for given adsorption energy.

Interestingly, some of the materials with sites in the $l$(C–O) > 1.30 Å subgroups were studied as supports for metallic nanoparticles. For instance, Ni/$LaAlO_3$ is a catalyst for dry reforming of methane[45] at 700 °C. It was shown that its catalytic performance is higher in terms of $CO_2$ and $CH_4$ conversion rates compared to Ni/$La_2O_3$ and Ni/$Al_2O_3$[45]. All sites on considered lanthanum (III) oxide surfaces belong to the subgroup of OCO < 132° without Sabatier constraint, whereas the sites on $Al_2O_3$ do not enter any of the two subgroups. $KNbO_3$ has been studied only with Pt nanoparticles and as a composite with g-$C_3N_4$ in photocatalytic reduction of $CO_2$ into $CH_4$[46,47]. Pt-$KNbO_3$ is ~2.5 times more photoactive than Pt-$NaNbO_3$[46], whereas the $NaNbO_3$ is known to be photoactive even without nanoparticles[48]. This seems to suggest that $l$(C–O) is a good indicator of $CO_2$ activation for both unsupported and supported catalysts even at increased temperatures. Hence, the other materials with the sites from this subgroup are promising new

**Table 3 The catalytic performance of materials which contain the sites from larger $l(C-O)$ or/and smaller OCO subgroups.**

| Material | Catalytic reaction | $CO_2$ adsorption energies, eV | Belong to subgroups |
|---|---|---|---|
| $NaNbO_3$ | Photocatalytic $CO_2$ reduction with ~70% of CO selectivity[46, 48] | −0.77 to −0.81 | Materials with sites from $l(C-O) > 1.30$ Å subgroup and OCO < 132° subgroup with Sabatier principle constraint |
| $LaAlO_3$ | Dry reforming of methane with Ni-nanoparticles; performance is higher than for Ni-$La_2O_3$ and Ni-$Al_2O_3$[45] | −1.17 | |
| $KNbO_3$ | Photocatalytic reduction of $CO_2$ into $CH_4$ as a composite with Pt/g-$C_3N_4$; significant improvement of activity when compared to Pt/g-$C_3N_4$; Pt-$KNbO_3$ is ~2.5 times more photoactive than Pt-$NaNbO_3$[46, 47] | −0.56 to −0.68 | |
| $CaTiO_3$ | $CO_2$ hydrogenation under UV-irradiation, although activity is not very high[51, 57]; twice higher activity with Ni-nanoparticles[57] | up to −2.70 | Materials with sites from $l(C-O) > 1.30$ Å subgroups and from OCO < 132° subgroup without Sabatier principle constraint |
| $CaZrO_3$, $SrZrO_3$, $BaZrO_3$, $SrTiO_3$ | Reverse water gas-shift reaction (RWGS) under 700–1100 °C[49] | up to −2.75 | |
| $SrTiO_3$ | Photocatalytic $CO_2$ methanation with Pt, Au-nanoparticles, significant decrease of activity during reaction[50] | up to −2.40 | |
| $YInO_3$[a] | No activity observed in photocatalytic $CO_2$ conversion[52] | −1.16−−1.47 | Materials with sites only from OCO < 132° subgroup without Sabatier principle constraint |
| CaO, SrO, BaO, $Na_2O$ | Strong carbonation, candidate materials for carbon capture and storage (CCS)[44] | −1.60 to −3.57 | |
| $La_2O_3$ | Dry reforming of methane with supported Ni-nanoparticles; lower performance than on Ni-$LaAlO_3$[45] and on some other supported catalysts[54] at 700 and 250 °C correspondingly | −2.14 to −3.11 | |
| CaO | Twice smaller reaction rate in $CO_2$ reforming of methane reaction with supported Ni-nanoparticles than on Ni-$La_2O_3$[58] at 750 °C | −1.60 to −3.42 | |
| $Ga_2O_3$ | Electrochemical reduction of $CO_2$ to formic acid[59]; (photo)catalytic hydrogenation of $CO_2$[60] | −0.74 to −1.34 | Materials with sites from OCO < 132° subgroup with Sabatier principle constraint |
| $Al_2O_3$ | Dry reforming of methane with supported Ni-nanoparticles[61]; lower performance than on Ni-$LaAlO_3$[45] | −0.87 | |

[a]Materials with sites also from OCO < 132° subgroup with Sabatier principle constraint.

candidates for this task. The most promising materials identified in this work are $CsNbO_3$, $CsVO_3$, $RbVO_3$, $LaScO_3$, $RbNbO_3$, and $NaSbO_3$ as they have the sites from the larger $l(C-O)$ subgroups satisfying the above-mentioned criteria.

There is also a set of materials [ternaries $A^{2+}B^{4+}O_3$ ($A$ = Ca, Sr, Ba, $B$ = Zr, Ti, Ge, Sn, Si) with a perovskite structure] containing both the surfaces with sites from the smaller OCO subgroups without Sabatier constraint and the surfaces with sites from the larger $l(C-O)$ subgroups (Table 3). These two types of sites are located on different surfaces. Thus, based on the above results, a material for which a surface with sites from the $l(C-O) > 1.30$ Å subgroups has lower formation energy and is more abundant than the surface with sites from smaller OCO subgroups without Sabatier constraint is expected to be a good catalyst. To explore this possibility, we analyze the surfaces of these materials in more detail. Their most stable surfaces are $AO$-terminated (001) facets containing sites from the smaller OCO subgroup. The formation energies of $ABO_3$-terminated (110) surfaces with larger $l(C-O)$ sites are higher: for $BaZrO_3$, $SrZrO_3$, $CaZrO_3$, and $SrTiO_3$ the differences in formation energies are 0.049, 0.027, 0.013, and 0.037 eV/Å², respectively. The zirconates and $SrTiO_3$ were found to catalyze the water gas-shift reaction under increased temperatures, 700–1100 °C[49]. At room temperature the photocatalytic activity of $SrTiO_3$ was found to be significantly decreased[50]. We attribute the latter finding to the strong carbonation of its most stable surface, which is consistent with the calculated high absolute value of $CO_2$ adsorption energy (−2.4 eV) for this surface. Thus, the activity

of $SrTiO_3$ at 700 °C and higher temperatures is consistent with the estimates of the $CO_2$ chemical potential given above. The difference in formation energies of the most stable CaO-terminated (001) surface and the stoichiometric (110) surface for $CaTiO_3$ is less pronounced compared to zirconates and other titanates (CaO-terminated (001) is more stable than the (110) surface by only 0.009 eV/Å²). Thus, the (110) facets, which contain sites from the long $l(C-O)$ subgroup, may be present on catalyst particles at the reaction conditions. This can explain the observed activity of $CaTiO_3$ in $CO_2$ conversion not only at high but also at room temperature. We note that the activity of this material was also attributed to the presence of $TiO_2$ nanoparticles on the surface[51] at reaction conditions.

The OCO subgroup that includes most of the known good catalysts and a minimal number of inactive materials is OCO < 132° with Sabatier principle. It contains the sites on discussed above $LaAlO_3$, $KNbO_3$, and $NaNbO_3$ catalysts, but also on non-active $YInO_3$ according to ref. [52] (Table 3). This subgroup contains in addition the sites on a well-known $CO_2$ conversion catalyst $Ga_2O_3$. We should mention that the catalytic activity of $Ga_2O_3$ has been attributed to its reducibility. According to Pan and coworkers[53] $CO_2$ molecules are activated via dissociation on surface O-vacancies. However, in ref. [54] only one $Ga_2O_3$ (100) surface was considered for which no energetically stable $CO_2$ chemisorption structures were obtained with the PBE functional. We show in Supplementary Table 1 and Supplementary Fig. 1 that this functional underestimates $CO_2$ adsorption energies. Moreover, in our study we considered also other surfaces and

found stable $CO_2$ chemisorption structures on these surfaces. Thus, activation of $CO_2$ on $Ga_2O_3$ can indeed proceed on O-atoms as discussed in our study, even without surface O-vacancies. The subgroups with small OCO cutoffs, 123° and 124°, do not contain any sites on known active or non-active catalysts.

OCO < 132° subgroup with Sabatier principle contains a large number of sites with elongated C–O bonds. The overlap of this subgroup with $l$(C–O) > 1.30 Å subgroups is 19 samples (70% of the latter).

To demonstrate the advantages of SGD over DTR in finding materials *genes* and their optimal combinations, we have done a comparison of found SGD subgroups with DTR performance for $l$(C–O). DTR terminal nodes (leaves) with the largest average $l$(C–O) (Supplementary Figs. 2 and 3) include surface sites on materials prone to extremely strong carbonation (Table 2), and also sites at which $CO_2$ prefers to physisorb, with $l$(C–O) = 1.17 Å. Also, one cannot check the effect of imposing the constraint as there is no standard way to mix regression and classification in DTR. Thus, DTR in contrast to SGD is not able to separate different activation modes and even fails sometimes in distinguishing activation from non-activation.

**Best materials for $CO_2$ reduction among calculated ones**. Now those good indicators of activation are identified (OCO with Sabatier principle and $l$(C–O)), all calculated materials can be ranked according to the value of these indicators (smaller OCO or larger $l$(C–O) indicate C–O bond weakening and therefore higher catalytic activity, provided adsorption energy is moderate). The resulting list of the most promising catalysts for $CO_2$ conversion is presented in Table 4. Each surface is characterized by maximum $l$(C–O) and minimum OCO among all inequivalent sites on that surface. The materials with $l$(C–O) > 1.30 Å are listed in the order of decreasing $l$(C–O). Materials with OCO < 132° but $l$(C–O) < 1.30 Å are appended at the bottom of the list in the order of increasing OCO.

Materials and surface cuts higher up in the list in Table 4 that belong to both $l$(C–O) > 1.30 Å and OCO < 132° subgroups are the most promising catalysts, followed by materials that belong to one of the subgroups, with the performance decreasing further down the list. Taking into account the number of active surface cuts and Sabatier principle, we conclude that $NaSbO_3$ is the most promising unexplored catalyst for temperatures up to 340 °C (for $CO_2$ pressures around 1 atm). Other $A^{+1}B^{+5}O_3$ type promising materials are $KSbO_3$ (for temperatures up to 110 °C) and $RbNbO_3$ (up to 360 °C) that belong to both subgroups, and $LiSbO_3$ (230 °C), $CsNbO_3$ (260 °C), $CsVO_3$ (110 °C), $NaVO_3$ (130 °C), belonging to one of the subgroups (listed in the order of decreasing performance). There are also several promising $A^{+3}B^{+3}O_3$ oxides with surfaces belonging to one or both subgroups, listed in the order they appear first time in the table: $ScAlO_3$ (up to 550 °C), $GaAlO_3$ (230 °C), $GaInO_3$ (340 °C), rhombohedral $InAlO_3$ (120 °C)—these and other In-containing materials are of course very expensive, but we list them here for completeness, $LaGaO_3$ (210 °C), $ScGaO_3$ (240 °C), $YAlO_3$ (330 °C).

From Table 4 it can be seen that not all promising materials belong to one of the found subgroups. This means that there are other optimal materials gene combinations that are not identified by SGD as statistically significant based on the current dataset. Such combinations may be unique for a given material, or they may be found when more data for different materials are considered. Among these materials the most promising are: $InScO_3$ (up to 430 °C), $MgSnO_3$ (430 °C), $CaGeO_3$ (570 °C), orthorhombic $InAlO_3$ (230 °C), $CaSiO_3$ (420 °C), $SrSiO_3$ (460 °C), $SrGeO_3$ (480 °C), and $BaSnO_3$ (up to 550 °C).

## Discussion

We have developed the subgroup-discovery strategy for finding improved oxide-based catalysts for the conversion of chemically inert molecules such as $CO_2$ into useful chemicals or fuels. For this purpose we identified a new indicator of $CO_2$ activation, namely the large C–O bond distance of the adsorbed molecule. This artificial-intelligence approach identifies the materials *genes* that correlate most strongly with the activation of the adsorbed molecule. Specifically, these are the following clean surface properties: Hirshfeld charges of O-atom at which $CO_2$ adsorbs ($q_O$) and of surface cations ($q_{min}$, $q_{max}$), surface geometric features [coordination descriptors $Q_i$, $i = 5$, 6, distances between the surface O-atom and the nearest surface cations ($d_i$, $i = 1$–3)], electrostatic potential and electric field above the adsorption site ($\Delta\varphi$, $\varphi_{2.6}$), polarizability and $C_6$ coefficients for surface atoms ($C_6^{min}$, $C_6^O$, $\alpha_{max}$), radii of HOMO and LUMO of the cation species ($r_{+1}^{max}$, $r_{+1}^{min}$, $r_{HOMO}^{min}$), ionization potential, electron affinity, and electronegativity of surface cation species ($IP_{max}$, $EA_{max}$, $EN_{min}$), features of O 2$p$ DOS ($kurt$, $M$, $PC$, $U$), conduction band minimum ($CBM$), energies of the lowest unoccupied projected eigenstates of surface cation species ($L_{max}$, $L_{min}$), and surface work function ($W$). The found subgroup selectors predict whether a given candidate material belongs to the class of promising catalysts. The peculiarity of the large C–O bond indicator is that it automatically satisfies Sabatier principle for low and middle-temperature $CO_2$ conversion.

The present study shows also that the previously proposed indicator for $CO_2$ activation, the decrease of the OCO-angle[27], is not appropriate and even correlates with strong adsorption so that poisoning by carbonation is likely which may be useful for carbon capture and storage (CCS) but not for carbon capture and utilization (CCU). When Sabatier principle is purposely included in the SGD search for small OCO, found subgroups substantially overlap with large $l$(C–O) subgroups (70%), although still contain a few sites on inactive materials for $CO_2$ conversion.

The subgroup analysis revealed an alternative mechanism of $CO_2$ activation by adsorption, namely bonding of an O-atom in $CO_2$ with a surface cation(s), combined with only moderate electron transfer from the surface to the molecule, which results not only in reduction of OCO-angles, but also in pronounced elongation and weakening of the C–O bond. Although the latter can be achieved also by a larger charge transfer, it results in stronger binding of $CO_2$ molecule to the surface and poisoning of the catalyst, contrary to the new mechanism. The same new mechanism is revealed when Sabatier principle is included when searching for small OCO subgroups.

We also demonstrated that a standard regression technique (DTR), which gives prediction models in a physically interpretable form similar to subgroup discovery (selectors based on identified descriptor), fails to identify the optimal combinations of materials *genes* and the activation in general. This failure is traced back to the fact that DTR is a global approach, which minimizes error in the prediction of the value of a target property for the whole dataset. As a result, different combinations of *genes* leading to the optimal value of the same target property are intermixed, and the combination that leads to the most optimal value is not identified. On the contrary, subgroup discovery finds unique local subsets in the data independent of the rest of the data. This makes it more suitable for identifying different combinations of materials *genes* that result in activation.

The other four considered potential indicators (charge at the adsorbed $CO_2$, adsorption induced dipole moment, the difference of charges on O-atoms and on C and O-atoms of adsorbed $CO_2$) were found to reproduce the results of SGD obtained for OCO-angles or C–O bond distances with significant overlap with corresponding subgroups.

**Table 4 Best materials and surface cuts for $CO_2$ activation according to the $l(C-O)$ and OCO indicators.**

| Material | Surface cut | $l(C-O)$, Å | OCO, degree | $E_{ads}$, eV | In $l(C-O) > 1.30$ Å subgroup | In OCO < 132° c. subgroup |
|---|---|---|---|---|---|---|
| According to $l(C-O)$ indicator | | | | | | |
| $NaSbO_3$ | 100 | 1.370 | 125.21 | −1.32 | Yes | Yes |
| $Ga_2O_3$ | 212 | 1.365 | 124.57 | −1.34 | | Yes |
| $NaSbO_3$ | 010 | 1.365 | 125.95 | −1.09 | Yes | Yes |
| $LiSbO_3$ | 010 | 1.359 | 126.66 | −1.04 | | Yes |
| $NaNbO_3$ | 100 | 1.353 | 125.87 | −0.78 | Yes | Yes |
| $ScAlO_3$ | 010 | 1.351 | 127.25 | −1.18 | | Yes |
| $KSbO_3$ | 110 | 1.345 | 128.54 | −0.72 | Yes | Yes |
| $LiNbO_3$ | 100 | 1.344 | 126.23 | −0.87 | | |
| $NaNbO_3$ | 010 | 1.344 | 126.85 | −0.77 | Yes | Yes |
| $InScO_3$ | 121 | 1.342 | 126.26 | −1.23 | | |
| $CsNbO_3$ | 100 | 1.34 | 126.6 | −0.87 | Yes | |
| $RbNbO_3$ | 111 | 1.338 | 126.61 | −1.37 | Yes | Yes |
| $CsNbO_3$ | 010 | 1.336 | 126.23 | −1.11 | Yes | |
| $MgSnO_3$ | 100 | 1.334 | 119.84 | −1.58 | | |
| $GaAlO_3$ | 100 | 1.332 | 129.12 | −1.02 | | Yes |
| $CaGeO_3$ | 001($GeO_2$-term.) | 1.331 | 127.65 | −0.75 | | |
| $InAlO_3$-or. | 121 | 1.33 | 130.09 | −1.02 | | |
| $ScAlO_3$ | 121 | 1.328 | 131.61 | −0.86 | | |
| $GaInO_3$ | 110 | 1.327 | 126.98 | −1.34 | Yes | |
| $LaAlO_3$ | 110 | 1.327 | 129.38 | −1.17 | Yes | Yes |
| $CsVO_3$ | 110 | 1.327 | 126.1 | −0.72 | Yes | |
| $KNbO_3$ | 110 | 1.327 | 128.49 | −0.68 | Yes | Yes |
| $RbVO_3$ | 110 | 1.326 | 126.04 | −1.14 | | |
| $Ga_2O_3$ | 110 | 1.325 | 127.76 | −1.09 | | Yes |
| $NaVO_3$ | 110 | 1.324 | 127.12 | −0.755 | Yes | |
| $NaNbO_3$ | 110 | 1.322 | 128.14 | −0.805 | Yes | Yes |
| $InAlO_3$-rh. | 110 | 1.318 | 126.83 | −0.73 | Yes | Yes |
| $LaGaO_3$ | 100 | 1.317 | 125.29 | −0.97 | Yes | |
| $ScGaO_3$ | 010 | 1.314 | 124.68 | −1.06 | | Yes |
| $GaInO_3$ | 120 | 1.313 | 118.41 | −1.43 | Yes | Yes |
| $MgGeO_3$-tetr. | 001($GeO_2$-term.) | 1.312 | 126.18 | −1.35 | | |
| $ScAlO_3$ | 100 | 1.312 | 122.28 | −1.89 | | Yes |
| $YAlO_3$ | 011 | 1.312 | 127.26 | −1.18 | Yes | Yes |
| $InScO_3$ | 110 | 1.31 | 122.28 | −1.54 | | Yes |
| $In_2O_3$ | 111 | 1.309 | 128.44 | −0.65 | | |
| $InAlO_3$-or. | 110 | 1.309 | 127.2 | −0.66 | | Yes |
| $YAlO_3$ | 100 | 1.308 | 123.82 | −1.305 | Yes | Yes |
| $InScO_3$ | 110($In_2O_3$-term.) | 1.305 | 124.92 | −1.57 | | Yes |
| $YGaO_3$ | 100 | 1.305 | 124.76 | −1.23 | | |
| $In_2O_3$ | 110 | 1.301 | 125.86 | −1.00 | | |
| $Sc_2O_3$ | 111 | 1.301 | 130.43 | −0.885 | | |
| $LaGaO_3$ | 110 | 1.301 | 128.88 | −0.83 | Yes | Yes |
| $LaScO_3$ | 100 | 1.301 | 123.6 | −1.53 | Yes | |
| according to OCO indicator | | | | | | |
| $CaSiO_3$ | 001(CaO-term.) | 1.290 | 118.84 | −1.54 | | |
| $SrSiO_3$ | 001(SrO-term.) | 1.295 | 119.10 | −1.66 | | |
| $CaGeO_3$ | 001(CaO-term.) | 1.288 | 120.88 | −1.94 | | |
| $Ga_2O_3$ | 212 | 1.297 | 121.21 | −1.53 | | |
| $InScO_3$ | 110 | 1.292 | 121.23 | −1.88 | | |
| $InScO_3$ | 100 | 1.277 | 121.40 | −1.74 | | |
| $RbVO_3$ | 100 | 1.283 | 121.64 | −0.53 | | |
| $In_2O_3$ | 110 | 1.280 | 122.52 | −1.57 | | |
| $InScO_3$ | 110($In_2O_3$-term.) | 1.284 | 122.80 | −1.78 | | |
| $SrGeO_3$ | 100(SrO-term.) | 1.277 | 122.90 | −1.70 | | |
| $TiO_2$-rutile | 100 | 1.276 | 123.61 | −1.05 | | |
| $ZrO_2$ | 111 | 1.280 | 123.72 | −0.92 | | |
| $BaSnO_3$ | 001(BaO-term.) | 1.267 | 123.80 | −1.89 | | |
| $ScGaO_3$ | 110 | 1.292 | 123.85 | −1.22 | | |
| $ZrO_2$ | 011 | 1.264 | 124.06 | −0.72 | | |
| $LiVO_3$ | 110 | 1.295 | 124.76 | −0.70 | | |
| $NaNbO_3$ | 010 | 1.273 | 125.00 | −1.66 | | |
| $MgTiO_3$ | 012 | 1.295 | 125.16 | −1.47 | | |
| $InAlO_3$-or. | 010 | 1.284 | 125.30 | −0.82 | | Yes |
| $YInO_3$ | 100 | 1.293 | 125.69 | −1.47 | | |
| $KNbO_3$ | 010 | 1.277 | 125.97 | −1.52 | | |

**Table 4 (continued)**

| Material | Surface cut | $l$(C–O), Å | OCO, degree | $E_{ads}$, eV | In $l$(C–O) > 1.30 Å subgroup | In OCO < 132° c. subgroup |
|---|---|---|---|---|---|---|
| InAlO$_3$-or. | 110 | 1.278 | 126.04 | −0.90 | | |
| ScAlO$_3$ | 110 | 1.277 | 126.10 | −1.33 | | |
| Al$_2$O$_3$ | 012 | 1.265 | 126.46 | −0.87 | | Yes |
| Sc$_2$O$_3$ | 110 | 1.265 | 126.47 | −1.14 | | |
| CaSiO$_3$ | 110(CaO-term.) | 1.278 | 126.49 | −1.44 | | |
| LaInO$_3$ | 100 | 1.287 | 127.13 | −1.27 | | |
| Sc$_2$O$_3$ | 111 | 1.265 | 127.49 | −0.95 | | |
| YInO$_3$ | 110 | 1.298 | 127.61 | −1.22 | | Yes |
| ScAlO$_3$ | 121 | 1.268 | 127.73 | −0.755 | | |
| MgTiO$_3$ | 001 | 1.265 | 127.85 | −1.37 | | |
| BaGeO$_3$ | 001(BaO-term.) | 1.270 | 128.50 | −1.80 | | |
| SrTiO$_3$ | 001(TiO$_2$-term.) | 1.266 | 128.53 | −1.92 | | |
| ZnO | 10–10 | 1.270 | 128.60 | −1.005 | | |
| YGaO$_3$ | 110 | 1.263 | 128.68 | −1.60 | | |
| SrSnO$_3$ | 001(SnO$_2$-term.) | 1.273 | 128.90 | −1.64 | | |
| Sc$_2$O$_3$ | 001 | 1.289 | 128.90 | −1.70 | | |
| MgGeO$_3$ | 001 | 1.260 | 128.93 | −1.09 | | |
| CaO | 001 | 1.262 | 129.20 | −1.60 | | |
| Al$_2$O$_3$ | 001 | 1.283 | 129.22 | −1.315 | | |
| BaSnO$_3$ | 001(SnO$_2$-term.) | 1.270 | 129.50 | −1.87 | | |
| CaSnO$_3$ | 001(SnO$_2$-term.) | 1.272 | 130.09 | −1.32 | | |
| KVO$_3$ | 010 | 1.267 | 130.17 | −0.55 | | |
| CaZrO$_3$ | 101(ZrO$_2$-term.) | 1.265 | 130.36 | −1.86 | | |
| CaSnO$_3$ | 110(SnO$_2$-term.) | 1.272 | 130.50 | −1.44 | | |
| SrGeO$_3$ | 100(GeO$_2$-term.) | 1.270 | 130.90 | −1.515 | | |
| CaTiO$_3$ | 101(TiO$_2$-term.) | 1.266 | 131.42 | −1.505 | | |
| SnO$_2$ | 100 | 1.257 | 131.50 | −0.85 | | |
| BaSiO$_3$ | 100 | 1.243 | 131.60 | −0.75 | | |
| MgO | 111 | 1.296 | 131.70 | −1.24 | | |

Based on our results, we propose several new promising oxide-based catalysts for $CO_2$ conversion (Table 4). Although the present work has focused on oxides only, the overall strategy is general and can be applied to any other family of materials. This work also emphasizes the importance of documenting metadata and workflows for AI data analysis in materials science in order to ensure the reproducibility of AI models and data analysis results.

## Methods

**Ab initio calculations**. The calculations are performed using density-functional theory (DFT) with the PBEsol exchange-correlation functional[55] as implemented in FHI-aims code[56] using 'tight' basis sets. The functional is chosen based on a comparison of calculated bulk lattice constants[55] and $CO_2$ adsorption energy to the available experimental results and high-level calculations (CCSD(T) and validated hybrid); see Supporting Information (SI) for more details on the computational setup. Nevertheless, it is expected that, because of the large set of systems inspected and the small variations introduced by the functional choice, the main trends will hold even when using another functional.

**Studied materials**. The dataset includes 71 semiconductor oxide materials, with 141 surfaces. The materials are ternary ($ABO_3$) and binary oxides with metal cations $A$ and $B$ from groups 1–5 (including La) and groups 12–15 of the periodic table. The full list of materials and surface cuts is given in Supplementary Notes, and the dataset is available in ref. [26]. In this study we considered only stoichiometric surface reconstructions obtained by atomic relaxation of stoichiometric bulk-like initial surface geometries. While this seems to be a limitation, our results show that indicators of activation calculated with this assumption correlate with experimental activity for known good oxide catalysts. This does not imply that surfaces of these materials do not reconstruct, but that the properties of unreconstructed surfaces can be used as descriptors for catalysis at reconstructed and defected surfaces under realistic conditions. The inclusion of surface reconstructions in the training data will further improve the predictions and will be a subject of future work.

**The details of SGD**. The SGD was done with the RealKD code (https://bitbucket.org/realKD/), modified to include quality functions described by Eqs. (1) and (2) in

which the information gain was defined as:

$$u(p) = 1 - \left(\frac{-1}{\ln 2}\right)(p \cdot \ln(p) + (1-p) \cdot \ln(1-p)) \quad (3)$$

here $p$ is the number of samples in a subgroup within the required adsorption energy range divided by the total number of samples in the subgroup. Since Shannon entropy is a symmetric parabola-like function around 0.5, we set here $F(Z) = 0$ for $p \le 0.5$. Also, $x \cdot \ln(x) = 0$ for $x = 0$. The search of subgroups is performed using a Monte-Carlo scheme adapted for these tasks[34].

The cutoff values $x$, $y$, ... used for setting propositions (feature-1 < $x$, feature-2 ≥ $y$, etc.) are obtained by $k$-means clustering, as implemented within RealKD. That is, for a desired number $n = k - 1$ of cutoff values a set of $k$ representative values of a given feature and $k$ groups (clusters) of the data points are determined that minimize the deviation of all the feature values from the representative values. Thus, each value of the feature in the dataset is assigned to a particular cluster, and the cutoffs are determined as the arithmetic mean between the closest feature values in neighboring clusters. The number $k$ is a parameter, and different $k$-values can in principle result in different cutoff values. It is worth noting that, due to the stochastic Monte-Carlo sampling, the exact definitions of the subgroups may vary for consecutive runs of the SGD algorithm. We have tested $k = 12$, 14, and 16 and rerun the algorithm several times for each $k$. While the results indeed depend on the run and on the $k$ value, the subgroups maximizing the quality function have largely or entirely overlapping populations, and selectors with the same or similar propositions. Here we report selectors that appear most often and have high population and quality function values.

**Decision-tree regression**. The DTR analysis was performed using Python scikit-learn libraries. DTR is a supervised learning method in which the training set is repeatedly split into patterns (so-called leaves) by means of propositions built from primary features. The fitting of a model is done with respect to the cost function, which encloses the deviation of fitted values of a target property from the actual values. In this study we considered two cost functions—mean squared error (MSE) and mean absolute error (MAE). The search for the most optimal partitioning (the so-called tree) is done with the greedy algorithm. To obtain the most optimal TR model, we used a standard approach for supervised machine learning—leave-one-out cross-validation with respect to the hyperparameters—minimal size of a leaf, maximal depth. The minimal size of a leaf is a bottom threshold of the population of a pattern, since too small size might result in overfitting. Maximal depth is a limit for the maximal number of splits in a tree.

## Data availability

The dataset is available in the NOMAD AI Toolkit[26].

## Code availability

A Jupyter notebook is available in the NOMAD AI Toolkit[26].

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

## Acknowledgements

We thank Mario Boley for fruitful discussions on SGD and for providing the RealKD (for SGD) code. We also thank Yoshi Tateyama and Xinyi Lin for helping to generate the bulk oxide models and Helena Muñoz Galan and Oriol Lamiel Garcia for preliminary calculations. This project has received funding from the European Union's Horizon 2020 research and innovation program (#951786: The NOMAD European Center of Excellence and the ERC grant #740233: TEC1p), the Spanish MICIUN/FEDER RTI2018-095460-B-I00 and *María de Maeztu* MDM-2017-0767 grants and, in part, by *Generalitat de Catalunya* 2017SGR13 grant, plus a generous allocation of computational time provided by the *Red Española de Supercomputación*—RES (QCM-2017-3-0006, QCM-2017-2-0005, QCM-2016-3-0005, QCM-2016-2-0007), and was supported by FAIRmat (FAIR Data Infrastructure for Condensed-Matter Physics and the Chemical Physics of Solids), DFG #460197019. The development of SGD approach was supported by Russian Science Foundation under grant 21-13-00419.

## Author contributions

M.S. and F.I. suggested the specific scientific problem and the general idea on methodology, A.M., Y.W., R.V., and F.V. generated the dataset, S.V.L. developed SGD methodology and modified the RealKD code, A.M. applied AI methodology to analyze the data, A.M., S.V.L., L.M.G., and M.S. interpreted the results, A.M., L.M.G., S.V.L., and M.S. established the Jupyter notebook, A.M., S.V.L., and L.M.G. wrote the manuscript.

## Funding

## Competing interests

The authors declare no competing interests.
