## [Peer Review File · Nature Communications]

Artificial-intelligence-driven discovery of catalyst *genes* with application to CO₂ activation on semiconductor oxidesREVIEWER COMMENTS

Reviewer #1 (Remarks to the Author):

In their manuscript, Mazheika et al. use subgroup discovery to find features in a material with a heightened weakening of the CO bond in CO₂, among other grandiose claims. What they say to be "AI" in this work is not AI, but simple data mining/clustering, with many of the conclusions being either meaningless or misleading for the actual purpose of designing better CO₂ activation materials. The analysis in the paper betrays a serious lack of understanding of surface chemistry and catalysis, and I cannot imagine anyone in the field can make use of this information in the paper. To put it nicely, this work is not publishable in Nature Communications in any shape or form, or even a specialized journal in this domain, no matter what fancy terminology and eye-catching titles are used.

Below are a non-exhaustive list of some specific issues:

-the introduction is confusing, introduces many new and unnecessary terminology, and skips over much of the existing literature on catalytic descriptors, including studies specific to the topic of CO₂ conversion.

-What the authors call "materials genes" would be referred to as catalytic descriptors in the vast majority of the existing literature, and this is far from being either a new concept or worth giving a new type of terminology

-the assertion in pg 8 that good CO₂ activation catalysts have CO₂ adsorption energies of -0.5 to -2 eV is simply strange and I have never seen or heard such a claim before, nor do the authors cite any similar references. the fact that the authors do not distinguish different types of CO₂ participating catalytic processes (electrocatalytic reduction, photocatalytic reduction, dry reforming, etc) in making this assumption is also illogical and inappropriate.

-the comparison between large I(CO) features and experimental catalytic activity in table 3 is largely a meaningless exercise due to the many other factors in governing activity; correlation here is not causation. furthermore, it is probably just as likely to find good catalysts with small I(CO) features. this is not rigorous science nor can any conclusion be drawn from it, and it is surprising to even see this attempted here.

-ultimately it is not clear a large I(CO) or OCO is even a good indicator of low CO₂ activation barriers, and the authors have not computed a single reaction barrier in this study. a large number of the "good" materials listed in table 4 have high CO₂ activation barriers; for example MgO and other similar oxides are simply not good at CO₂ activation.

Reviewer #2 (Remarks to the Author):

The manuscript deals with finding descriptors for oxides to activate CO₂. While there are some important developments here I find missing a deep analysis and comparison to previous works in the literature.

Particularly, I find missing references to the search of descriptors for oxides developed by the groups of Prof. Pérez-Ramírez and Lopez on the subject that presented the earlier analysis by Grasselli in this field. Other recent oxide descriptors have been derived by M. Andersen and Reuter

and Linic's group and also deserve proper comparison.

Experimental results show that CO₂ conversion requires also to activate a second compound, most commonly H₂. The previous works of Prof. Pérez-Ramírez on CO₂ conversion need to be referred properly too.

Reviewer #3 (Remarks to the Author):

The authors present the results of a thought-provoking study using AI methods to discover new oxide-based materials for discovering new catalysts for CO₂ conversion. This paper has many merits such as: the inclusion of the Sabatier principle in their analysis; is the use of a comprehensive list of potential indicators of reactivity to create subgroups (with different choices creating subgroups with strong overlap); rigorous documentation of the metadata in an accessible notebook; an approach which is extensible to other materials/reactions,; and a thoughtful analysis of the findings . The paper is well written and presented and in general is a solid representation of how Ai could be used in catalyst discovery.

That said I do have one major concern, validation of findings. My worry is that AI predictions of improved materials are becoming ubiquitous with the quality of the AI work showing larger variations from paper to paper. This current manuscript, in my opinion, is clearly from the technical end on the stronger end of the spectrum but ultimately it needs to make CCURATE predictions about new materials. I appreciate that the groups naturally cluster together CO₂ capture materials and discover known reactive materials with potentially new ones but even the results of table 4 bring together a lot of materials (some of which lie outside the best subgroups). However, to me the study would be significantly increased in its value if it can produce concrete and validatable findings. Without containing any experimental studies to validate this discovery the paper must depend on follow on work by other groups and for that I feel this ensemble of materials is still quite broad. With such a broad group of materials the likelihood of successfully identifying a new one that is active towards Co₂ is likely, but it is not clear to me what is the probability of a false positive.? Can there be further organized into groups which would be good of some reaction conditions and potentially be prioritized in their likelihood. I think a more detailed discussion on validation of findings is needed and perhaps even a more focused prediction.

Overall, this paper has great potential to be an archival achievement in AI and I hope the authors find the above suggestion useful.

Reply to referees

Referee 1:

"In their manuscript, Mazheika et al. use subgroup discovery to find features in a material with a heightened weakening of the CO bond in CO₂, among other grandiose claims. What they say to be "AI" in this work is not AI, but simple data mining/clustering, with many of the conclusions being either meaningless or misleading for the actual purpose of designing better CO₂ activation materials. The analysis in the paper betrays a serious lack of understanding of surface chemistry and catalysis, and I cannot imagine anyone in the field can make use of this information in the paper. To put it nicely, this work is not publishable in Nature Communications in any shape or form, or even a specialized journal in this domain, no matter what fancy terminology and eye-catching titles are used."

Reply: We warmly recommend the referee to improve their understanding of the AI discipline, as their comment is an obvious proof that the referee lacks some basic knowledge in the field. SGD is a data mining approach that expresses the results in human-readable language. According to the classic book (Principles of Data Mining by David Hand, Heikki Mannila and Padhraic Smyth ISBN: 026208290x The MIT Press © 2001): "Data mining is often set in the broader context of knowledge discovery in databases, or KDD. This term originated in the artificial intelligence (AI) research field." See also Ertel, Wolfgang. Introduction to artificial intelligence. Springer, 2018, where data mining is a chapter. In general, data mining is often called foundation of AI. Regarding our "lack" of understanding of catalysis, the referee's general claims are not supported by the list of specific issues, as explained below, and are therefore just unsupported claims.

"the introduction is confusing, introduces many new and unnecessary terminology, and skips over much of the existing literature on catalytic descriptors, including studies specific to the topic of CO₂ conversion."

Reply: These comments are very vague, no specific examples of missed relevant studies are given. We are not writing a general review, and citing all literature on descriptors and CO₂ activation would not help to convey the specific messages in our paper. Therefore, we cite and discuss only the most relevant studies.

"What the authors call "materials genes" would be referred to as catalytic descriptors in the vast majority of the existing literature, and this is far from being either a new concept or worth giving a new type of terminology"

Reply: In our definition, materials genes is a more general concept than descriptors. The "genes" do not necessarily correlate with catalytic activity by themselves. Similarly to biological genes, their role depends on the combination in which they occur, and can be either beneficial or detrimental to the catalytic activity. We add this explanation to the text to make the term clearer:

P.3. The "genes" do not necessarily correlate with catalytic activity by themselves. Similarly to biological genes, their role depends on the combination in which they occur, and can be either beneficial or detrimental to the catalytic activity.

“the assertion in pg 8 that good CO₂ activation catalysts have CO₂ adsorption energies of -0.5 to -2 eV is simply strange and I have never seen or heard such a claim before, nor do the authors cite any similar references. the fact that the authors do not distinguish different types of CO₂ participating catalytic processes (electrocatalytic reduction, photocatalytic reduction, dry reforming, etc) in making this assumption is also illogical and inappropriate.”

Reply: Is the referee familiar with the Sabatier principle, which is one of the central concepts in catalysis? Regretfully, this comment is in line with many publications that completely neglect this key principle and report “good catalysts” that either bind the reactants too strongly or do not bind them at all at realistic conditions. As pointed out by another referee, taking into account the Sabatier principle is in fact one of the strengths of our work. And yes, this principle is relevant for any type of catalysis. Regarding the particular selected adsorption energy range, we give a very clear physical reasoning for its choice.

“the comparison between large I(CO) features and experimental catalytic activity in table 3 is largely a meaningless exercise due to the many other factors in governing activity; correlation here is not causation. furthermore, it is probably just as likely to find good catalysts with small I(CO) features. this is not rigorous science nor can any conclusion be drawn from it, and it is surprising to even see this attempted here.”

Reply: When we use I(CO) as an indicator of activation, we get subgroups that include most of the known good catalysts within the considered class of materials. We do this using the rigorous data analysis. Using I(CO) is based on very clear physical principle (bond elongation → weakening). In addition, we use other indicators of activation, also based on clear physical grounds, and we get consistent results. Thus, saying that “it is probably just as likely to find good catalysts with small I(CO) features” contradicts the available data for the considered class of materials.

“ultimately it is not clear a large I(CO) or OCO is even a good indicator of low CO₂ activation barriers, and the authors have not computed a single reaction barrier in this study. a large number of the “good” materials listed in table 4 have high CO₂ activation barriers; for example MgO and other similar oxides are simply not good at CO₂ activation.”

Reply: We say very clearly that CO₂ needs to be activated and that activation is the first and crucial step. See also ref. [Freund, H.-J. & Roberts M. W. Surface chemistry of carbon dioxide. Surf. Sci. Rep. 25, 225-273 (1996)]. Obviously, the full catalytic process is affected by more processes and there is intricate interplay. Regarding MgO, we note that in addition to the material composition it is important to distinguish different surface cuts, as is done in Table 4. The MgO surface that is predicted to be active (although with a border value of the activation indicator and not entering any of the subgroups) is (111). This is a polar surface which is much less thermodynamically stable than the inactive (100) in a wide range of conditions. We find this result very interesting, because it means that MgO could activate CO₂, if the surfaces are properly engineered. In fact there is a study reporting enhanced carbon capture at MgO(111) nanosheets (DOI:10.1021/jacs.8b01845). We are not aware of similar studies on catalytic conversion of CO₂, and we see this as another reason to publish our paper in Nature Comm..

To clarify which materials we consider as promising, we added the following text:

P. 20. Materials and surface cuts higher up in the list in Table 4 that belong to both $l(\text{C-O}) > 1.30 \text{ \AA}$ and $\text{OCO} < 132^\circ$ subgroups are the most promising catalysts, followed by materials that belong to one of the subgroups, with the performance decreasing further down the list. Taking into account the number of active surface cuts and Sabatier principle, we conclude that NaSbO_3 is the most promising unexplored catalyst for temperatures up to $340 \text{ }^\circ\text{C}$ (for CO_2 pressures around 1 atm). Other $A^{+1}B^{+5}\text{O}_3$ type promising materials are KSbO_3 (for temperatures up to $110 \text{ }^\circ\text{C}$) and RbNbO_3 (up to $360 \text{ }^\circ\text{C}$) that belong to both subgroups, and LiSbO_3 ($230 \text{ }^\circ\text{C}$), CsNbO_3 ($260 \text{ }^\circ\text{C}$), CsVO_3 ($110 \text{ }^\circ\text{C}$), NaVO_3 ($130 \text{ }^\circ\text{C}$), belonging to one of the subgroups (listed in the order of decreasing performance). There are also several promising $A^{+3}B^{+3}\text{O}_3$ oxides with surfaces belonging to one or both subgroups, listed in the order they appear first time in the table: ScAlO_3 (up to $550 \text{ }^\circ\text{C}$), GaAlO_3 ($230 \text{ }^\circ\text{C}$), GaInO_3 ($340 \text{ }^\circ\text{C}$), rhombohedral InAlO_3 ($120 \text{ }^\circ\text{C}$) – these and other In-containing materials are of course very expensive, but we list them here for completeness, LaGaO_3 ($210 \text{ }^\circ\text{C}$), ScGaO_3 ($240 \text{ }^\circ\text{C}$), YAlO_3 ($330 \text{ }^\circ\text{C}$).

P. 23. Among these materials the most promising are: InScO_3 (up to $430 \text{ }^\circ\text{C}$), MgSnO_3 ($430 \text{ }^\circ\text{C}$), CaGeO_3 ($570 \text{ }^\circ\text{C}$), orthorhombic InAlO_3 ($230 \text{ }^\circ\text{C}$), CaSiO_3 ($420 \text{ }^\circ\text{C}$), SrSiO_3 ($460 \text{ }^\circ\text{C}$), SrGeO_3 ($480 \text{ }^\circ\text{C}$), and BaSnO_3 (up to $550 \text{ }^\circ\text{C}$).

Referee 2:

“The manuscript deals with finding descriptors for oxides to activate CO_2 . While there are some important developments here I find missing a deep analysis and comparison to previous works in the literature.”

Reply: We thank the referee for the comments.

“Particularly, I find missing references to the search of descriptors for oxides developed by the groups of Prof. Pérez-Ramírez and Lopez on the subject that presented the earlier analysis by Grasselli in this field. Other recent oxide descriptors have been derived by M. Andersen and Reuter and Linic's group and also deserve proper comparison.”

Reply: We have added the following discussion to the main text:

P. 7.: Our strategy is to provide an almost exhaustive list of features, and use data analytics to select materials genes from this list. Some of these features have been explored previously as descriptors of catalytic activity for oxides.^{38,39,40,41} O 2p band center features have been shown to correlate with catalytic properties of both semiconducting and metallic oxides^{38,40}. In particular, most of the features (or closely related ones) mentioned in Ref. 39, inspired by the work of Grasselli⁴², are included in our set, except oxygen vacancy formation energy, which is relevant for the oxidation catalysis, while here we are interested in partial or complete reduction. Additional important features in our work (see below) include features related to polarizability of surface cations, which describe long-range surface response to charged adsorbates. A subset of features from our list have been recently used successfully for predicting catalytic properties of metallic oxides⁴¹, along with additional features relevant specifically for metallic oxides (such as partial electronic state fillings).

We note that an earlier version of our work with the same list of features was published on arxiv [<https://arxiv.org/abs/1912.06515v1>] before the publications by Andersen and Reuter and Linic's group on the relevant topic.

“Experimental results show that CO₂ conversion requires also to activate a second compound, most commonly H₂. The previous works of Prof. Pérez-Ramírez on CO₂ conversion need to be referred properly too.”

Reply: We thank the referee for pointing this important issue. We have now cited the works of Prof. Pérez-Ramírez and some other works properly, and added the following sentences to clarify the issue with source of activated hydrogen:

P. 4. Catalytic conversion of CO₂ requires activation of other reactants as well, e.g. molecular hydrogen, water, or methane. In particular hydrogen can serve as an environmentally friendly reagent that can be produced by water electrolysis or photo-splitting avoiding extra CO₂ emissions.²²⁻²⁴ Also, oxygen vacancies have been proposed as active sites for CO₂ conversion on some materials.²⁵ Therefore, predictions of catalytic activity of materials for CO₂ conversion can be refined based on analysis of activation of other reactants and defects. Nevertheless, as shown below, meaningful predictions can be made based solely on the analysis of adsorption properties of CO₂ on pristine surface. This confirms that these properties are good indicators of activation with a viable optimization pathway at least for the chosen class of materials.

Referee 3:

“The authors present the results of a thought-provoking study using AI methods to discover new oxide-based materials for discovering new catalysts for CO₂ conversion. This paper has many merits such as: the inclusion of the Sabatier principle in their analysis; is the use of a comprehensive list of potential indicators of reactivity to create subgroups (with different choices creating subgroups with strong overlap); rigorous documentation of the metadata in an accessible notebook; an approach which is extensible to other materials/reactions; and a thoughtful analysis of the findings. The paper is well written and presented and in general is a solid representation of how AI could be used in catalyst discovery.

That said I do have one major concern, validation of findings. My worry is that AI predictions of improved materials are becoming ubiquitous with the quality of the AI work showing larger variations from paper to paper. This current manuscript, in my opinion, is clearly from the technical end on the stronger end of the spectrum but ultimately it needs to make CCURATE predictions about new materials. I appreciate that the groups naturally cluster together CO₂ capture materials and discover known reactive materials with potentially new ones but even the results of table 4 bring together a lot of materials (some of which lie outside the best subgroups). However, to me the study would be significantly increased in its value if it can produce concrete and validatable findings. Without containing any experimental studies to validate this discovery the paper must depend on follow on work by other groups and for that I feel this ensemble of materials is still quite broad. With such a broad group of materials the likelihood of successfully identifying a new one that is active towards CO₂ is likely, but it is not clear to me what is the probability of a false positive.? Can there be further organized into groups which would be good of some reaction conditions and potentially be prioritized in their likelihood. I think a more detailed discussion on validation of findings is needed and perhaps even a more focused prediction.”

Reply: We thank the referee for the positive assessment of our work and constructive criticism. We have updated the paper to address it:

P. 20. Materials and surface cuts higher up in the list in Table 4 that belong to both $l(\text{C-O}) > 1.30 \text{ \AA}$ and $\text{OCO} < 132^\circ$ subgroups are the most promising catalysts, followed by materials that belong to one of the subgroups, with the performance decreasing further down the list. Taking into account the number of active surface cuts and Sabatier principle, we conclude that NaSbO_3 is the most promising unexplored catalyst for temperatures up to $340 \text{ }^\circ\text{C}$ (for CO_2 pressures around 1 atm). Other $A^{+1}B^{+5}\text{O}_3$ type promising materials are KSbO_3 and RbNbO_3 that belong to both subgroups, and LiSbO_3 , CsNbO_3 , CsVO_3 , NaVO_3 , belonging to one of the subgroups (listed in the order of decreasing performance). There are also several promising $A^{+3}B^{+3}\text{O}_3$ oxides with surfaces belonging to one or both subgroups, listed in the order they appear first time in the table: ScAlO_3 , GaAlO_3 , GaInO_3 , rhombohedral InAlO_3 (these and other In-containing materials are of course very expensive, but we list them here for completeness), LaGaO_3 , ScGaO_3 , YAlO_3 .

P. 23. Among these materials the most promising are: InScO_3 (up to $430 \text{ }^\circ\text{C}$), MgSnO_3 ($430 \text{ }^\circ\text{C}$), CaGeO_3 ($570 \text{ }^\circ\text{C}$), orthorhombic InAlO_3 ($230 \text{ }^\circ\text{C}$), CaSiO_3 ($420 \text{ }^\circ\text{C}$), SrSiO_3 ($460 \text{ }^\circ\text{C}$), SrGeO_3 ($480 \text{ }^\circ\text{C}$), and BaSnO_3 (up to $550 \text{ }^\circ\text{C}$).

REVIEWERS' COMMENTS

Reviewer #1 (Remarks to the Author):

The paper remains deeply flawed in its analysis and of limited use in guiding the discovery of CO₂ activation materials. The authors have made some attempts at a rebuttal. I have no further comments or suggestions.

Reviewer #2 (Remarks to the Author):

The authors have considered in a very mild manner (still issues with citations) however, I will not prevent the manuscript for being published.

Reviewer #3 (Remarks to the Author):

The revisions are adequate